# Use of Hyperspectral Reflectance and Water Quality Indices to Assess Groundwater Quality for Drinking in Arid Regions, Saudi Arabia

**Abdulaziz Alqarawy** [1,2]**, Maged El Osta** [1,]*****, **Milad Masoud** [1]**, Salah Elsayed** [3] **and Mohamed Gad** [4]

1   Water Research Center, King Abdulaziz University, Jeddah 21589, Saudi Arabia;
    aalqaraawi@kau.edu.sa (A.A.); mhmasoud@kau.edu.sa (M.M.)
2   Faculty of Meteorology, Environment and Arid Land Agriculture, King Abdulaziz University,
    Jeddah 21589, Saudi Arabia
3   Agricultural Engineering, Evaluation of Natural Resources Department, Environmental Studies and Research
    Institute, University of Sadat City, Sadat City 32897, Egypt; salah.emam@esri.usc.edu.eg
4   Hydrogeology, Evaluation of Natural Resources Department, Environmental Studies and Research Institute,
    University of Sadat City, Sadat City 32897, Egypt; mohamed.gad@esri.usc.edu.eg
*   Correspondence: melosta@kau.edu.sa

**Abstract:** Combining hydrogeochemical characterization and a hyperspectral reflectance measurement can provide knowledge for groundwater security under different conditions. In this study, comprehensive examinations of 173 groundwater samples were carried out in Makkah Al-Mukarramah Province, Saudi Arabia. Physicochemical parameters, water quality indices (WQIs), and spectral reflectance indices (SRIs) were combined to investigate water quality and controlling factors using multivariate modeling techniques, such as partial least-square regression (PLSR) and principal component regression (PCR). To measure water quality status, the drinking water quality index (DWQI), total dissolved solids (TDS), heavy metal index (HPI), contamination degree ($C_d$), and pollution index (PI) were calculated. Standard analytical methods were used to assess nineteen physicochemical parameters. The typical values of ions and metals were as follows: $Na^{2+} > Ca^{2+} > Mg^{2+} > K^+$, $Cl^- > SO_4^{2-} > HCO_3^- > NO_3^- > CO_3^{2-}$; and $Cu > Fe > Al > Zn > Mn > Ni$, respectively. The hydrogeochemical characteristics of the examined groundwater samples revealed that $Ca$-$HCO_3$, $Na$-$Cl$, mixed $Ca$-$Mg$-$Cl$-$SO_4$, and $Na$-$Ca$-$HCO_3$ were the main mechanisms governing groundwater chemistry and quality under the load of seawater intrusion, weathering, and water-rock interaction. According to the WQIs results, the DWQI values revealed that 2.5% of groundwater samples were categorized as excellent, 18.0% as good, 28.0% as poor, 21.5% as extremely poor, and 30.0% as unfit for drinking. The HPI and $C_d$ values revealed that all groundwater samples had a low degree of contamination and better quality. Furthermore, the PI values showed that the groundwater resources were not affected by metals but were slightly affected by Mn in Wadi Fatimah due to rock–water interaction. Linear regression models demonstrated the significant relationships for the majority of SRIs paired with DWQI (R varied from −0.40 to 0.75), and with TDS (R varied from 0.46 to 0.74) for the studied wadies. In general, the PLSR and PCR models provide better estimations for DWQI and TDS than the individual SRI. In conclusion, the grouping of WQIs, SRIs, PLSR, PCR, and GIS tools provides a clear image of groundwater suitability for drinking and its controlling elements.

**Keywords:** groundwater quality; water facies; geochemical processes; spectral reflectance indices; drinking water; pollution indices

## 1. Introduction

Groundwater is a fundamental and critical resource for humans all around the planet. Because of the importance of this resource, the world's nations face a major challenge of water scarcity, especially in arid and semiarid regions [1,2]. Low precipitation and high

evaporation rates in arid and semiarid regions increase water salinity, which increases the toxicity of some chemicals in groundwater [3]. According to recent studies, an estimated 2.1 billion people do not have access to adequate quality fresh water [4]. Water security that fulfils the quality specifications stipulated in relevant standards is currently the foundation for the operation of most communities [5].

Makkah Al-Mukarramah Province, located in southwest Saudi Arabia, has about 43 coastal wadies that are characterized by relatively heavy flashfloods and groundwater resources. It is a densely populated area due to its religious location, which leads to a sharp increase in water demand and affects the quality of its water. The groundwater in the Makkah Al-Mukarramah region is the major source of water resources, which have decreased significantly over the past few years. As the KSA Vision "2030" aims to maximize the utilization of Kingdom resources, such as water, and keep them from deteriorating in order to provide a decent life for its citizens, and with the ever-increasing population, the need for natural resources increases in order to secure the needs of the society. Therefore, periodic and seasonal monitoring of groundwater levels and quality has become extremely important for evaluating groundwater and overcoming the problem of depletion and degradation of groundwater in this region. The traditional methods for assessing groundwater quality from sample collection, conservation, and laboratory analysis have become very difficult and costly. Therefore, it was necessary to propose an alternative solution to the periodic chemical analysis of water depending on the hydrochemical models, spectral reflectance measurements, and remote sensing technology.

Water quality management, water pollution control, and environmental preservation must all be emphasized in order to safeguard living circumstances [6,7]. Understanding groundwater chemistry is a necessary prerequisite for mechanism analysis and quality evolution [8–10]. Natural processes such as dissolution, percolation, precipitation, and cation exchange, as well as anthropogenic activities, define the mechanism for groundwater chemistry [11–13].

In past investigations, standard water quality indices (WQIs) were used to assess groundwater quality [14,15]. Considering the numerous hydrochemical criteria, the WQIs technique is ineffective for determining groundwater quality [16]. Because of its more complete calculation, the methodology for evaluating groundwater quality using numerous hydrochemical parameters is seen to be a more robust approach [17–20]. Therefore, the WQIs have been extensively used for assessing groundwater quality. Furthermore, the geography information system (GIS) is useful in determining the spatial distribution of WQI values. Global research has been conducted in order to establish appropriate water quality assessment methodologies, including the single factor index approach based on water body objectives and needs [21,22].

The most frequent techniques for assessing water quality are the DWQI, HPI, $C_d$, and PI. As a result, it is vital to improve the method used to compute the weight, which might represent tacit information included in data [23]. The WQIs are derived from a large information set that includes multiple water quality metrics from diverse places. Many WQIs have been established to serve as markers for evaluating water quality in both drinking and agriculture usages [24–27]. The primary purpose of WQIs is to transform large amounts of complex information into numeric datasets, which helps to improve knowledge of water quality. The DWQI might be established as a dependable measure, defined as a number that indicates the integrated influence of several water quality elements. As a result, The DWQI is calculated by assessing the combined effect of man-made and environmental activity in the hydro-geometry characteristics of groundwater in the study area [28].

Remote sensing, in conjunction with geospatial approaches, plays an important role in measuring and mapping environmental contamination by facilitating spatial assessment. These strategies rely on non-spatial data being spatially represented in order to map the researched parameter throughout the entire region, including non-surveyed locations [29]. Moreover, spatial data have made tremendous advances in health and environmental appraisal research, assisting in the detection of pollutants and the concerning environmental

elements [30]. The integration of remote sensing and GIS, as well as field observations and laboratory experiments, have generated exceptional environmental visualization of data with the help of spatial variation maps, which facilitate strategic planning [31]. Many studies commonly employ the visible and near-infrared ranges of the sun spectrum to extract accurate connections between SRIs and single water quality metrics (SWQM). Total suspended solids, turbidity, and chlorophyll-a were typically assessed utilizing remote sensing monitoring equipment [32,33]. However, defining a single water quality standard based on water quality usage is problematic.

There is particularly limited information available on the efficacy of spectral indices for predicting drinking and irrigation indices for water. There is just one study that discovered how to assess the drinking water quality index for water using known models based on spectral reflectance measurements to classify water quality level for drinking [34]. Consequently, the purpose of this research was to assess the robustness of recently created and published SRIs for calculating groundwater water-quality indices in three chosen wadies (Marawani, Fatimah, and Qanunah) in the Makkah region. As a consequence, freshly generated spectrum indices coming from the two wavelengths in the spectra will be assessed by building correlation matrices (contour maps) based on linear, quadratic, and exponential equations. Instead of formulating indices, whole spectra were used on an empirical basis to best fit the model to predict the trait of interest. Partial least square regression (PLSR) and principal component regression (PCR) models are typical methods that define an empirical rationale for a fit of the accuracy model to predict traits of interest [35]. Because the characteristics may be examined throughout a wide range of wavelengths from the spectral regions VIS and NIR or based on spectral indices SRIs, the PLSR and PCR are regression techniques that may be used to reduce the amount of collinear spectral factors generated [36,37].

Our research focuses on assessing groundwater quality in three different wadies utilizing physiochemical parameters, WQIs, and the accuracy of proximal remote sensing based on spectral bands. The objectives of this study were to: (i) identify geochemical types and their controlling factors based on physical and chemical metrics; (ii) assess the suitability of groundwater for drinking in relation to WQIs; (iii) assess the performance of newly and published SRIs for estimating the four WQIs of DWQI, TDS, HPI, and $C_d$; and (iv) evaluate the efficacy of PLSR and PCR models depending on investigated SRIs to predicting the four WQIs.

By concentrating on the most useful water quality criteria, this research offers tools for improved decision making concerning groundwater quality assessment in arid regions to guarantee successful management, aids in the identification of contamination sources, and offers a clearer picture for the redesign of sampling methods.

## 2. Materials and Methods

### 2.1. Study Wadies Description

The administrative borders of the Makkah province are bounded by the Red Sea in the west, Riyadh in the east, Al-Madinah Al-Munawarah in the north, and the Provinces of Al Bahaand Asir in the south, with a total area of 141,216 km$^2$ (Figure 1). This work was carried out under different condition in three representative wadies in Makkah Al-Mukarramah Province to evaluate the groundwater quality for drinking (Figure 1). Topographically, the Makkah region is characterized by a great diversity in altitude ranging from 0.0 m to 2984.0 m above mean sea level (amsl), as shown in Figure 2a [38]. Geologically, the lithological units that dominated in this area belong to the era that extends from the pre-Cambrian to the Quaternary (Figure 2b) [39]. The average annual precipitation distribution map in the Makkah region was constructed (Figure 3). Wadi Marawani, Wadi Fatimah, and Wadi Qanunah received an average rainfall of about 70–110 mm, 50–110 mm, and 100–400 mm, respectively, due to the presence of the upper mountainous region of the wadies [40].

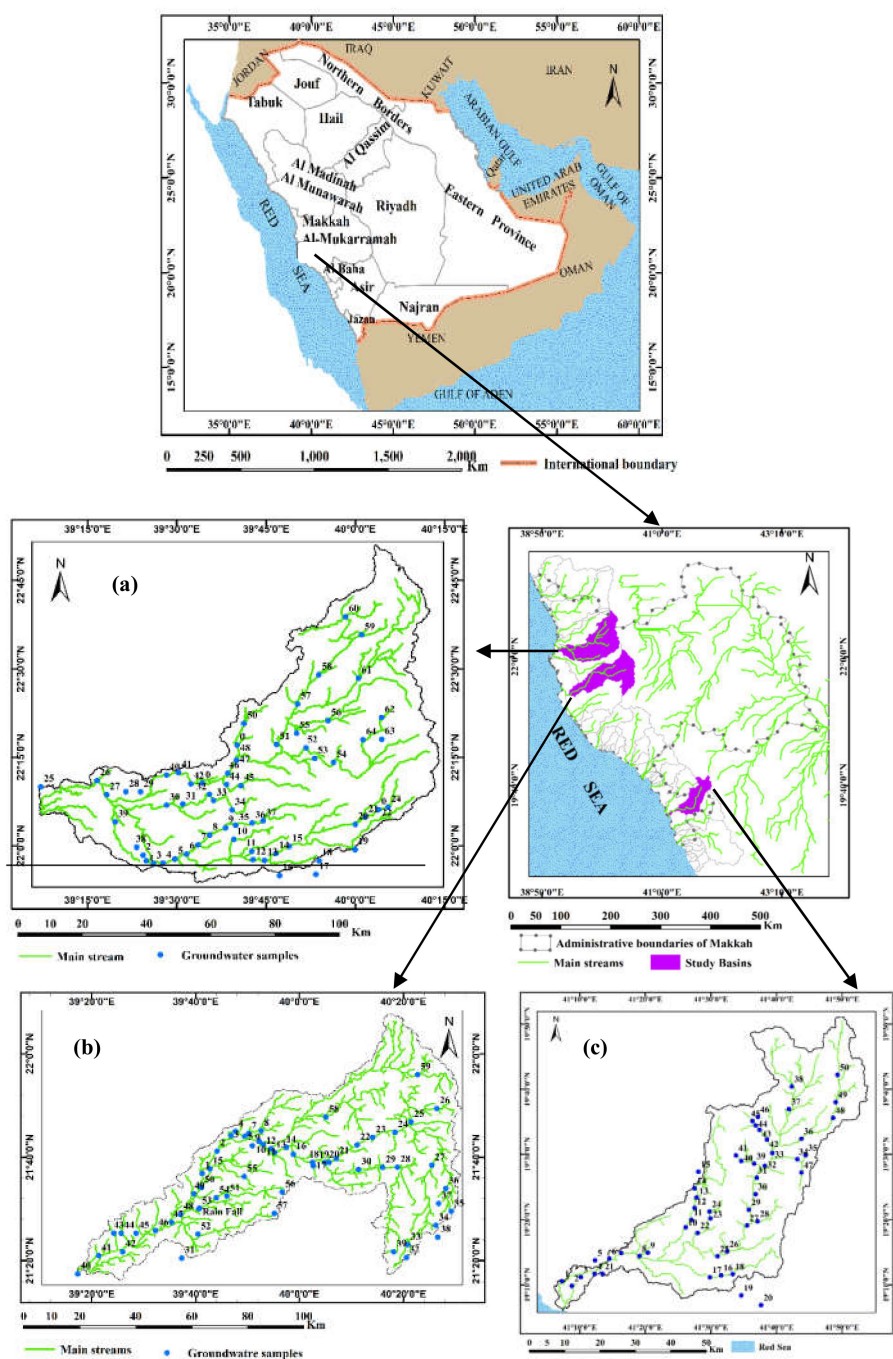

**Figure 1.** Location map of the groundwater points for the selected wadies in Makkah Al-Mukarramah Province, Saudi Arabia: (**a**) Wadi Marawani, (**b**) Wadi Fatimah, and (**c**) Wadi Qanunah.

Hydrogeological Description

The main water bearing formation in the selected wadies was the Quaternary aquifer, which is the primary source of groundwater for different uses. According to the information collected from the comprehensive survey of 173 groundwater wells, this aquifer was extended in coastal plains and wadi channels with a maximum thickness of about 118 m in Wadi Marawani, 60 m in Wadi Fatimah, and 74 m in Wad Qanunah. Lithologically, the Quaternary aquifer in the studied wadies in the Makkah region was formed mostly from alluvial deposits (gravel, sand, and silt) and the igneous metamorphic rocks form the bedrock of the aquifer, which were highly fractured and weather-cracked, making them an ideal host for groundwater preservation [40]. Therefore, the Quaternary aquifer

in the study area is of an unconfined condition. As a result of the shallow depth of the groundwater and the porous and permeable deposits on top of the aquifer, human and agricultural activities could contribute to pollution in these wadies. The amount of rainfall distributed along these waterways is a major source of groundwater recharge. The majority of rainfall runoff infiltrates into highly permeable sediments that fill the wadi channels, where it recharges the shallow underlying Quaternary aquifer.

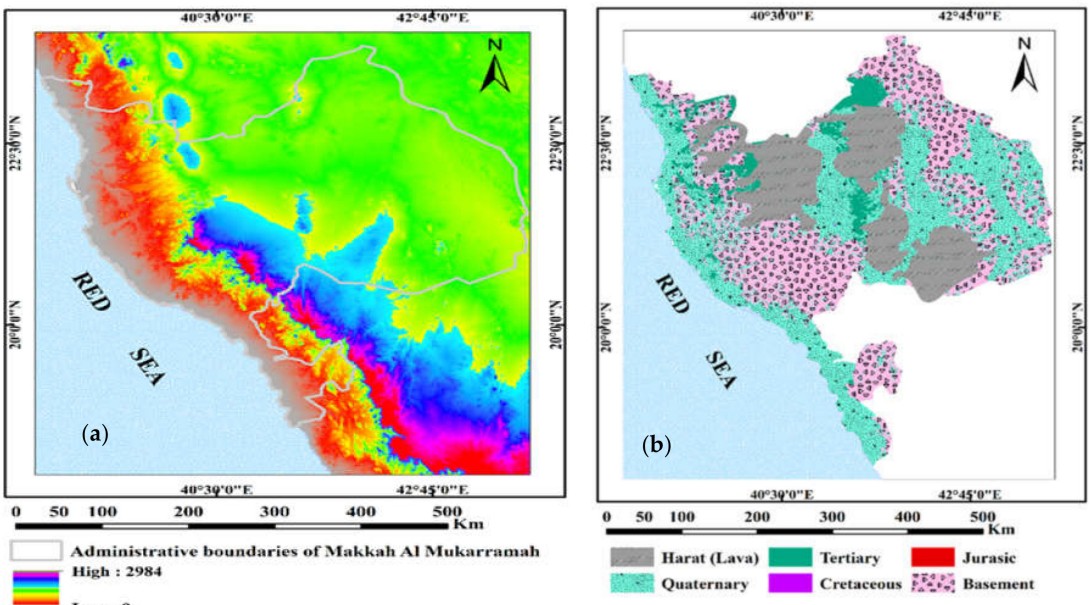

**Figure 2.** (**a**) Digital elevation model (DEM) and (**b**) geological map of Makkah Al-Mukarramah Province, Saudi Arabia.

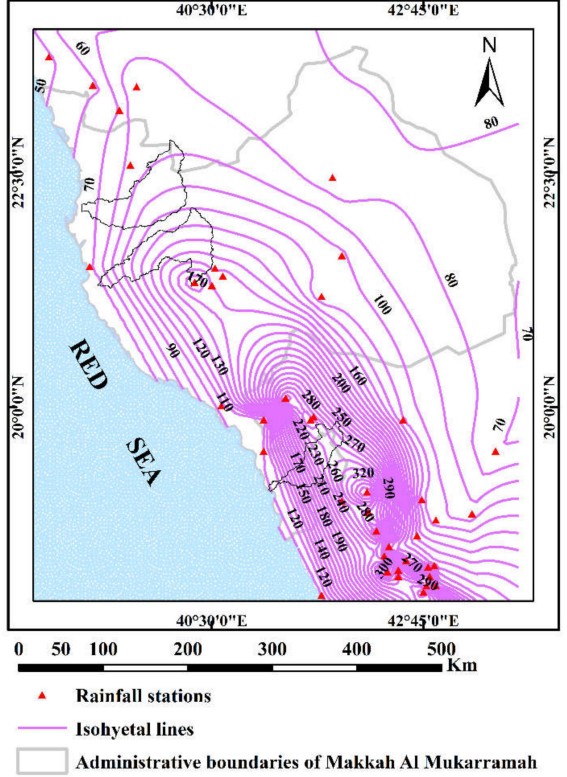

**Figure 3.** Annual distribution map of average rainfall (mm) in Makkah Al-Mukarramah Province, Saudi Arabia (1960–2012).

### 2.2. Groundwater Sampling

During 2021, groundwater samples were collected from the Quaternary aquifer along the three wadies. Sixty-four samples were taken from depths ranging between 1 and 30 m for Wadi Marawani, 59 samples from 1.2 to 50.1 m for Wadi Fatimah, and 50 samples between 0.8 and 21.7 m for Wadi Qanunah, through a surveying trip to convey the overall genuine conditions for assessing groundwater quality for drinking purposes. A portable GPS (MAGELLAN GPS 315, USA/United States of America) was used to determine the position of the water samples and identify the Universal Transverse Mercator coordinates system UTM of the research area (Figure 1).

### 2.3. Field Measurments and Laboratory Analysis

Portable conductivity multi-parameter equipment calibrated with standard solutions was used to determine T (°C), pH, EC, and TDS at the site (Hanna HI 9033, Germany). For chemical analysis, two groups of groundwater samples were obtained in 500 mL from each sampling location and filtered using 0.45 m Whatman filter paper. The samples were maintained at 4 °C until they were transported to the laboratory for physiochemical investigation. Using standard analytical procedures, twenty distinct physiochemical parameters were determined. The EDTA titrimetric approach was used to measure $Mg^{2+}$ and $Ca^{2+}$ concentrations, whereas a flame photometer was used to evaluate $K^+$ and $Na^{2+}$ concentrations (ELEX 6361, Hamburg, Germany). Titration with $AgNO_3$ was employed to measure $Cl^-$ concentrations, whereas titrimetry was utilized to detect $HCO_3^-$ and $CO_3^{2-}$ concentrations. Sulfate ($SO_4^{2-}$) and $NO_3^-$ concentrations were determined using a spectrophotometer with UV spectrum (DR/2040 Loveland, USA). For analysis of trace element, samples were adjusted to a pH of 2 with conc. $HNO_3$ preservatives before being analyzed. Standard analytical protocols [41] were utilized to analyze metals such as Al, Cu, Fe, Mn, Ni, and Zn with an inductively coupled plasma mass spectrometer (ICP-MS, Waltham, USA/United States of America). Several quality assurance methods were employed throughout the evaluation of the water samples. The charge balance errors (CBE) were determined after the laboratory double-checked the experimental data using the formula below (Equation (1)).

$$CBE = \frac{\sum Cations - \sum Anions}{\sum Cations + \sum Anions} \times 100 \qquad (1)$$

### 2.4. Measuring Spectral Reflectance

The spectral reflectance of the 173 collected groundwater samples from the three wadies was measured using a passive sensor (tec5, Oberursel, Germany). To study the link between the surface water spectrum in situ and the water quality indicators, a passive sensor with a viewing angle of 12° was utilized to measure the reflection of water. The water sample was placed in a glass jar, and the spectral reflectance of the water samples was measured at a distance of 0.30 m. A black sheet was used to cover the edges of the glass jar to avoid reflection from the backdrop and to collect all reflectance of water samples. The passive sensor has a spectral range from 302 nm to 1148 nm and a bandwidth of 2 nm [42]. The water reflection was obtained at a sunny period in a short time to eliminate variations in light radiation intensity, and each sample was a ten-scan average.

### 2.5. Selection of SRIs of Groundwater Samples of the Three Wadies

Fourteen SRIs, including five published indices and nine newly developed indices, were examined (Table 1). Contour maps were established to show determination coefficients ($R^2$) between the DWQI, TDS, HPI, and $C_d$ of water samples with ratio spectral indices (RSI). For example, the best RSI was determined by combining two separate wavelengths in the spectra ranging from 302 to 1148 nm for DWQI of three wadies: (a) Wadi Marawani, (b) Wadi Fatimah, and (c) Wadi Qanunah, as shown in Figure 4. The spectral reflectance contour maps of the spectrum region were constructed, which could be used to determine the effective spectral area with optimum wavelengths and to recognize the importance of

SRIs. SRIs were calculated using thirteen wavelengths (454, 470, 472, 480, 488, 510, 480, 510, 554, 570, 590, 1122, and 1124 nm).

**Table 1.** Description of different SRIs used in this work.

| SRIs | Formula | References |
|---|---|---|
| **Published SRIs** | | |
| Ratio between blue and red | Blue/Red | [43] |
| Ratio between green and red | Green/Red | [44] |
| Ratio between NIR and red | NIR/Red | [45] |
| Ratio between NIR and blue | NIR/Blue | [45] |
| Ratio between NIR and green | NIR/Green | [45] |
| New SRIs | | |
| $RSI_{1122,454}$ | $R_{1122}/R_{454}$ | This work |
| $RSI_{1122,470}$ | $R_{1122}/R_{470}$ | |
| $RSI_{1124,472}$ | $R_{1124}/R_{472}$ | |
| $RSI_{1122,480}$ | $R_{1122}/R_{480}$ | |
| $RSI_{1122,488}$ | $R_{1122}/R_{488}$ | |
| $RSI_{1122,510}$ | $R_{1122}/R_{510}$ | |
| $RSI_{1122,554}$ | $R_{1122}/R_{554}$ | |
| $RSI_{1124,570}$ | $R_{1122}/R_{570}$ | |
| $RSI_{1122,590}$ | $R_{1122}/R_{590}$ | |

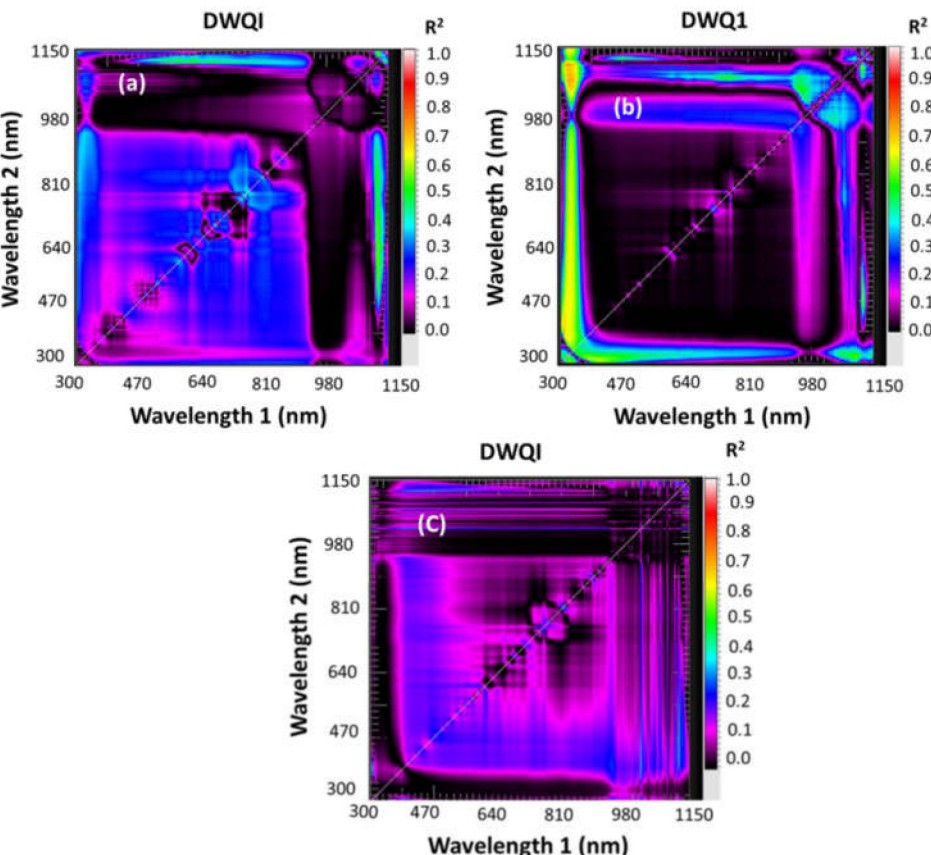

**Figure 4.** Correlation matrices displaying ($R^2$) values for possible dull wavelength ranging from 302 to 1148 nm with DWQI of three wadies: (**a**) Wadi Marawani, (**b**) Wadi Fatimah, and (**c**) Wadi Qanunah.

As indicated in Table 1, a ratio spectral index was calculated using this equation:

$$RSI = R_1/R_2 \qquad (2)$$

where $R_1$ and $R_2$ refer to the values of spectral reflectance at various wavelengths.

$$\text{The linear fitting equation: } Y = a + b \, RSI \tag{3}$$

### 2.6. Indexing Approach

The DWQI, HPI, $C_d$, and PI were calculated based on cited references [46–49] to detect the suitability of groundwater quality in the three selected wadies.

### 2.6.1. Drinking Water Quality Index (DWQI)

The DWQI were derived by the concentrations of thirteen determinants across the selected wadies (Table 2). The results of all laboratory analyses for all groundwater wells obtained were examined for quality review. The physicochemical parameters were weighted depending on their relevance in the overall quality of water for water supply. According to Equation (4), the DWQI displays the overall water quality of each water component as a function of a range of water quality factors and their utilization in the ecosystem [50].

$$\text{DWQI} = \sum_{i=1}^{n} W_i \times \left( \frac{C_i}{S_i} \times 100 \right) \tag{4}$$

where $W_i$ indicates the weight unit of each parameter, and 13 physical and chemical parameters were used. The concentration ($C_i$) and standard ($S_i$) values for each water variable were derived using the formula below (Equation (5)):

$$W_i = \frac{w_i}{\sum w_i} \tag{5}$$

**Table 2.** Arithmetic rating method for calculation of drinking water quality index (DWQI).

| Physicochemical Parameters | Measured Sample | $S_i$ (mg/L) WHO (2011) | Unit Weight $W_i$ | Sub Index ($Q_i$) | $Q_i \times W_i$ |
|---|---|---|---|---|---|
| pH | 7.70 | 8.5 | 0.4155 | 46.4162 | 19.2842 |
| EC | 3696.45 | 1500 | 0.0024 | 246.4297 | 0.5802 |
| TDS | 2214.00 | 500 | 0.0071 | 442.7992 | 3.1274 |
| TH | 1004.30 | 500 | 0.0071 | 200.8604 | 1.4187 |
| $K^+$ | 8.80 | 12 | 0.2943 | 73.3304 | 21.5802 |
| $Na^{2+}$ | 415.20 | 200 | 0.0177 | 207.5998 | 3.6656 |
| $Mg^{2+}$ | 89.34 | 50 | 0.0706 | 178.6869 | 12.6204 |
| $Ca^{2+}$ | 255.20 | 75 | 0.0471 | 340.2723 | 16.0220 |
| $Cl^-$ | 818.44 | 250 | 0.0141 | 327.3740 | 4.6244 |
| $SO_4^{2-}$ | 531.45 | 250 | 0.0141 | 212.5783 | 3.0028 |
| $HCO_3^-$ | 186.25 | 120 | 0.0294 | 155.2062 | 4.5675 |
| $CO_3^{2-}$ | 2.39 | 350 | 0.0101 | 0.6837 | 0.0069 |
| $NO_3^-$ | 46.83 | 50 | 0.0706 | 93.6599 | 6.6151 |
| | | | $\sum (W_i) = 1$ | | $\sum_{i=1}^{n} Q_i \times W_i$ |

Equation (6) computes $w_i$ for each parameter using the accepted criteria [51]:

$$w_i = K/S_i \tag{6}$$

where K denotes the proportionality factor.

To calculate the DWQI, each groundwater variable ($w_i$) is weighted, and the relative weight is established ($W_i$). Therefore, the values of $W_i$ were given to totally physicochemical parameters, and $w_i$ was computed using Equation (7). Table 2 displayed the calculated values of the water parameters' standards, weights ($w_i$), and arithmetic weights ($W_i$).

2.6.2. Water Pollution Indices (PIs)

Heavy Metal Pollution Index (HPI)

To determine total groundwater quality, a toxicity index (HPI) based on rating the mathematical weights of metals was utilized (Table 3). The HPI values represent the cumulative effect of elements on overall groundwater quality [52] in comparison to the suggested standard recommendations ($S_i$) for each metal, namely, Al, Cu, Fe, Mn, Ni, and Zn. The HPI values were calculated using Equation (7):

$$\text{HPI} = \frac{\sum_{i=1}^{n} W_i Q_i}{\sum_{i-1}^{n} W_i} \tag{7}$$

where $W_i$ and $Q_i$ are the corresponding unit weights and sub-indices for Al, Cu, Fe, Mn, Ni, and Zn, and n is the number of elements examined. There are three categories of HPI values: low heavy metal pollution (HPI > 100), heavy metal pollution with threshold risk (HPI = 100), and excessive heavy metal pollution (HPI > 100) [53].

**Table 3.** The HPI, $C_d$, and PI are computed using an arithmetic rating approach.

| Trace Element (mg/L) | Measured Sample | $S_i$ (mg/L) (WHO, 2011) | $MAC_i$ | Unit Weight $W_i$ | Sub Index $Q_i$ | $Q_i \times W_i$ |
|---|---|---|---|---|---|---|
| Al | 0.003 | 0.2 | 200 | 0.072 | 1.500 | 0.108 |
| Cu | 0.08 | 2 | 2000 | 0.007 | 4.000 | 0.029 |
| Fe | 0.016 | 0.3 | 300 | 0.048 | 5.333 | 0.257 |
| Mn | 0.002 | 0.1 | 100 | 0.145 | 2.000 | 0.289 |
| Ni | 0.014 | 0.02 | 20 | 0.723 | 70.000 | 50.602 |
| Zn | 0.006 | 3 | 3000 | 0.005 | 0.200 | 0.001 |
| | | | | $\sum (W_i) = 1$ | | $\sum_{i=1}^{n} Q_i \times W_i$ |

Contamination Index ($C_d$)

The contaminated components of certain heavy metals that were above allowable levels were used to compute groundwater contamination degrees, which are expressed by $C_d$ values [47,48] according to Equations (8) and (9):

$$C_d = \sum_{i=1}^{n} C_{fi} \tag{8}$$

$$C_{fi} = \frac{C_{Ai}}{C_{Ni}} - 1 \tag{9}$$

where $C_{fi}$ represents the contamination factor for each heavy metal, $C_{Ai}$ represents the analytical value for each metal, $C_{Ni}$ is the acceptable concentration of each element, and $C_{Ni}$ is reserved as the maximum acceptable concentration.

Pollution Index (PI)

To assess the influence of heavy metal contamination for the groundwater quality, PI values were calculated. These demonstrate the particular contaminating impact of each heavy metal on groundwater quality and are categorized into five groups (Table 4) based on Equation (10):

$$\text{PI} = \frac{\sqrt{[(\frac{C_i}{S_i})^2_{max} + (\frac{C_i}{S_i})^2_{min}]}}{2} \tag{10}$$

where $C_i$ denotes the metal content in water, and $S_i$ indicates the metal level based on standards for each metal [54].

**Table 4.** Pollution levels as determined by PI according to Edet and Offong [48].

| Class | PI Value | Effect |
|-------|----------|--------|
| 1 | <1 | No effect |
| 2 | 1–2 | Slightly affected |
| 3 | 2–3 | Moderately affected |
| 4 | 3–5 | Strongly affected |
| 5 | >5 | Seriously affected |

*2.7. Partial Least-Square Regression (PLSR) and Principal Component Regression (PCR)*

Both the PLSR and PCR models were utilized in this study to predict the DWQI, TDS, HPI, and $C_d$. Both models of the four WQIs were created using unscramble X software version 10.2. The two models (PLSR and PCR) included all selected SRIs in Table 1 as input datasets to forecast the four WQIs as output data for each wadi. The independent parameters were associated to the dependent parameter using leave-one-out cross-validation (LOOCV) for two models. The ideal number of latent variables (LVs) was selected based on the lowest value of RMSE to describe the calibration data without overfitting or underfitting (Figure 5). To strengthen the robustness of the results, the datasets were subjected to random 10-fold cross-validation.

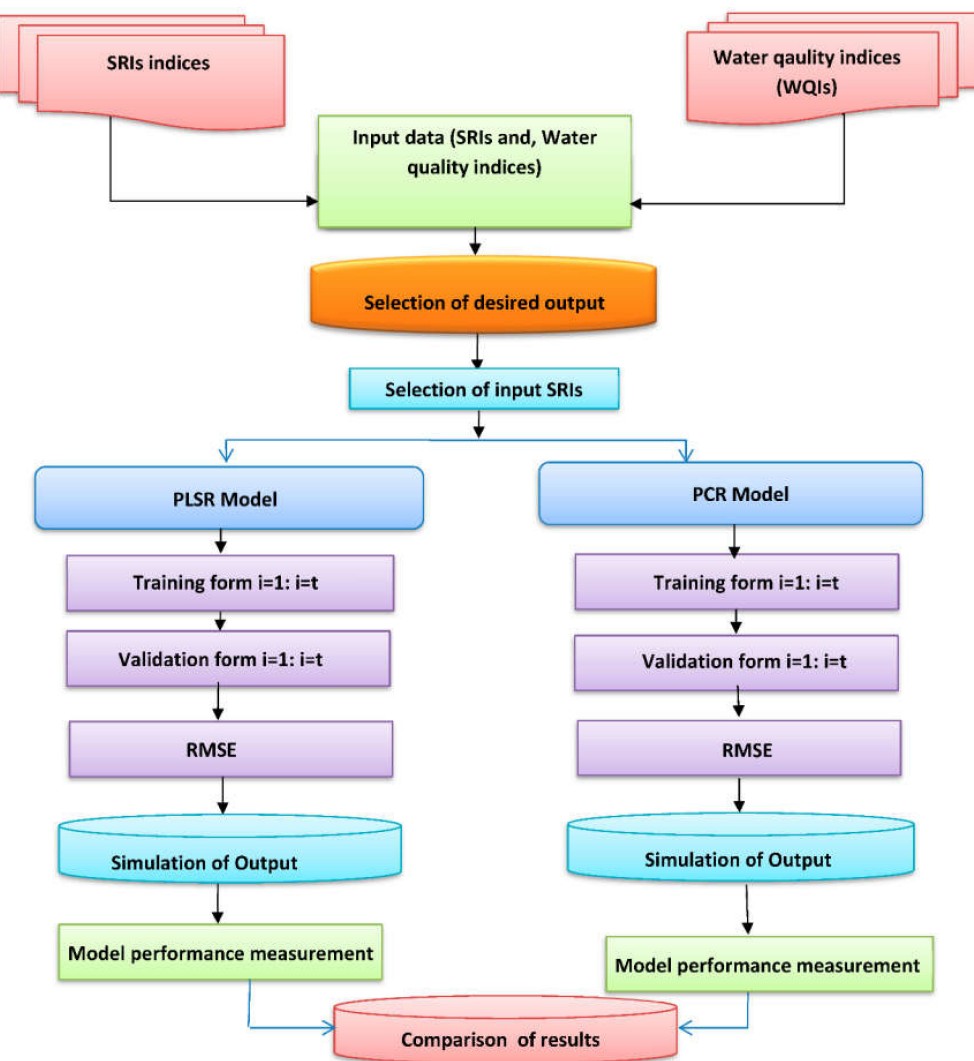

**Figure 5.** Schematic flowchart of the process of PLSR and PCR used to assess DWQI, TDS, HPI, and $C_d$ of water samples in this study.

Three metrics were employed to assess the two models' effectiveness in predicting the four WQIs: $R^2$ coefficient, RMSE, and equation slope.

$R^2$ coefficient:

$$R^2 = 1 - \frac{\sum_{i=1}^{n}(WQIso_i - WQIs_{fi})^2}{\sum_{i=1}^{n}(PIso_i)^2} \tag{11}$$

RMSE:

$$RMSE = \sqrt{\frac{\sum_{i=1}^{n}(WQIso_i - WQIs_{fi})^2}{n}} \tag{12}$$

where $WQIso_i$ is the measured value, n is the number of data points, and $WQIs_{fi}$ is the predicted value.

*2.8. Data Analysis and Graphical Approach*

The descriptive statistics, including the minimum, maximum, and mean of the physicochemical parameters, WQIs, and SRIs, were calculated using SPSS 22 (SPSS Inc., Chicago, IL, USA). Hydrochemical facies generation models, such as the Piper and Gibbs diagrams, have been developed to establish groundwater types, geochemistry processes, and primary chemical control variables utilizing Geochemist's Software package 12.0. In order to characterize the geochemical types of groundwater in the analyzed region, Piper's trilinear diagram was used [55] to quantify the predominant cations and anions in meq/L of the collected groundwater samples. Gibbs diagram is widely used to establish a connection between groundwater composition and the aquifer matrix [56]. Gibbs diagram identifies the fundamental regulatory mechanisms that influence groundwater geochemistry by presenting TDS vs. (Na + K)/(Na + K + Ca) and Cl/(Cl + HCO₃). GIS version 10.0 was used to construct spatial variation maps for WQIs tested in this study. In addition, a combination of geo-statistical data processing and physicochemical characteristics was used to construct the WQI maps.

## 3. Results and Discussion

### 3.1. Physicochemical Parameters

Physicochemical metrics are significant criteria for assessing the geochemistry of groundwater and related signaling pathways, and so play a significant role in water quality management. Table 5 provides statistical summaries (min., max., mean) of the physicochemical properties of the groundwater locations studied in the three selected wadies.

The data analysis of physicochemical properties for groundwater samples from three wadies revealed that groundwater temperatures varied from 23.0 to 32.0 °C, with an average of around 28.08 °C based on well depths to the water surface. The pH readings varied from 6.99 to 8.39, with a mean of 7.69, indicating that the groundwater was mildly alkaline. The EC measurements varied from 429.0 to 28,700 µS/cm, with a mean of 3696.44 µS/cm. The TDS values ranged from 208.0 to 18518 mg/L, with a mean value of 2213.99 mg/L, which indicates brackish to saline groundwater, especially for Marawani and Fatimah basins, which indicated the possibility of seawater intrusion, especially in wells near the coast, and with the increase for discharging from those wells (mixing process). The relatively fresh water of Wadi Qanunah was due to the high annual rainfall amount along this wadi (100–400 mm), where infiltration rates are sufficient to fill the alluvial deposits within a short amount of time, and, consequently, at places of high permeability, and the surface water depressions appear directly connected to shallow unconfined aquifers [40]. The ionic content of $K^+$, $Na^{2+}$, $Mg^{2+}$, $Ca^{2+}$, $Cl^-$, $SO_4^{2-}$, $HCO_3^-$, $CO_3^{2-}$, and $NO_3^-$ displayed mean values of 8.79, 415.19, 89.34, 415.19, 818.43, 531.44, 186.24, 2.39, and 46.82 mg/L, respectively (Table 5). Therefore, the average ion values displayed sequences of $Na^{2+} > Ca^{2+} > Mg^{2+} > K^+$, and $Cl^- > SO_4^{2-} > HCO_3^- > NO_3^- > CO_3^{2-}$, respectively. These values revealed that $Na^{2+}$ was the dominant cation and $Cl^-$ was the dominant anion for the collected water samples. The high concentration of $NO_3^-$ in groundwater samples was mostly caused by nitrate fertilizer leaching from agricultural areas and domestic wastew-

ater leakage from the nearby residential area [39]. The average concentrations of trace elements (Al, Cu, Fe, Mn, Ni, and Zn) across the selected wadies were 0.0239, 0.1316, 0.0300, 0.0204, 0.0095, and 0.0341 mg/L, respectively, which exhibited the following tendency: Cu > Fe > Al > Zn > Mn > Ni (Table 5). We know that trace elements in groundwater originate from a variety of sources, including weathering, soil leaching, and human activity. The quantities of trace elements in the analyzed groundwater samples showed low level of contamination by metals that were less than the specified permitted limits for drinking usages [50]. Accordingly, these elements were distributed uniformly, indicating that they were mainly derived from dissolution of silicate minerals in the aquifer matrix [39].

**Table 5.** Statistical analysis of the physical and chemical parameters of groundwater samples in the three selected wadies, Makkah Al-Mukarramah Province.

| Physicochemical Parameters | Wadi Marawani (n = 64) | | | Wadi Fatimah (n = 59) | | | Wadi Qanunah (n = 50) | | |
|---|---|---|---|---|---|---|---|---|---|
| | Min. | Max. | Mean | Min. | Max. | Mean | Min. | Max. | Mean |
| T °C | 24.00 | 31.00 | 27.30 | 30.00 | 32.00 | 30.66 | 23.00 | 30.00 | 26.04 |
| pH | 7.10 | 8.00 | 7.67 | 6.99 | 8.39 | 7.74 | 7.12 | 8.14 | 7.68 |
| EC | 658.00 | 28,700.00 | 4905.52 | 553.00 | 25,000.00 | 4217.27 | 429.00 | 4010.00 | 1534.26 |
| TDS | 346.00 | 18,171.00 | 2936.54 | 227.00 | 18,518.00 | 2572.27 | 208.00 | 2375.00 | 866.38 |
| TH | 67.18 | 7914.51 | 1298.81 | 44.17 | 6032.46 | 1189.46 | 136.31 | 1143.57 | 408.84 |
| $K^+$ | 0.79 | 28.10 | 8.12 | 0.99 | 79.03 | 13.87 | 0.60 | 13.57 | 3.69 |
| $Na^+$ | 38.00 | 5150.00 | 588.90 | 43.64 | 4602.76 | 441.93 | 24.10 | 641.40 | 161.32 |
| $Mg^{2-}$ | 9.30 | 710.00 | 129.67 | 4.12 | 575.28 | 90.56 | 9.00 | 133.40 | 36.28 |
| $Ca^{2+}$ | 11.60 | 2002.00 | 306.85 | 10.92 | 1995.81 | 327.26 | 34.50 | 415.00 | 104.08 |
| $Cl^-$ | 37.10 | 9666.00 | 1193.12 | 70.53 | 7271.04 | 926.22 | 14.70 | 901.50 | 211.65 |
| $SO_4{}^{2-}$ | 19.30 | 2840.00 | 609.62 | 30.01 | 5180.28 | 692.35 | 25.70 | 945.00 | 241.51 |
| $HCO_3{}^-$ | 31.00 | 394.00 | 200.50 | 12.20 | 274.50 | 146.19 | 104.00 | 356.00 | 215.28 |
| $CO_3{}^{2-}$ | N.D. | N.D. | N.D. | N.D. | 24.00 | 7.02 | N.D. | N.D. | N.D. |
| $NO_3{}^-$ | 2.20 | 290.70 | 53.99 | 0.01 | 475.44 | 57.28 | 0.78 | 160.80 | 25.34 |
| Al | 0.007 | 0.233 | 0.024 | 0.003 | 0.073 | 0.014 | 0.002 | 0.233 | 0.037 |
| Cu | 0.006 | 0.643 | 0.181 | 0.005 | 0.080 | 0.022 | 0.005 | 0.726 | 0.198 |
| Fe | 0.006 | 0.415 | 0.039 | 0.010 | 0.025 | 0.017 | 0.006 | 0.175 | 0.034 |
| Mn | 0.007 | 0.192 | 0.028 | 0.002 | 0.285 | 0.011 | 0.008 | 0.108 | 0.021 |
| Ni | 0.008 | 0.021 | 0.011 | 0.001 | 0.017 | 0.008 | 0.007 | 0.015 | 0.010 |
| Zn | 0.005 | 0.740 | 0.053 | 0.001 | 0.090 | 0.009 | 0.005 | 0.226 | 0.040 |

Note: Except for temperature (T °C), pH, and EC (μS/cm), all physical and chemical characteristics are provided in mg/L.

### 3.2. Geochemical Facies and Controlling Processes

Hydrogeochemical data analysis was conducted through illustrative methodologies, such as Piper and Gibbs diagrams, to better understand the different geochemical influences that control the groundwater quality in the studied wadies. According to the chemical characteristics of the groundwater samples analyzed, the hydrochemical facies were $Ca-HCO_3$, Na-Cl, mixed $Ca-Mg-Cl-SO_4$, and $Na-Ca-HCO_3$ (Figure 6). The weathering process and the aquifer matrix significantly alter the chemical composition of groundwater. A plot of chemical data on a Gibbs diagram (Figure 7) showed that the groundwater samples of the three wadies were distributed throughout the weathering and rock–water interactions fields [9].

### 3.3. Water Quality Indices

The statistical analysis and categorization of the various WQIs, including the DWQI, HPI, and $C_d$, employed in this study are shown in Table 6. Furthermore, spatial distribution maps for each index were utilized to present the quality of water in the researched basins (Table 6).

**Table 6.** Classification of several water quality indices (WQIs).

| WQIs | Min. | Max. | Mean | Range | Water Category | Number of Samples (%) | | |
|---|---|---|---|---|---|---|---|---|
| | | | | | | Wadi Marawani (n = 64) | Wadi Fatimah (n = 59) | Wadi Qanunah (n = 50) |
| DWQI | 22.69 | 545.53 | 97.11 | 0–25 | Excellent | 0 (0.0%) | 1 (2.0%) | 3 (6.0%) |
| | | | | 26–50 | Good | 7 (11.0%) | 2 (3.0%) | 22 (44.0%) |
| | | | | 51–75 | Poor | 24 (37.5%) | 10 (17.0%) | 15 (30.0%) |
| | | | | 76–100 | Very poor | 9 (14.0%) | 19 (32.0%) | 9 (18%) |
| | | | | >100 | Unsuitable | 24 (37.5%) | 27 (46.0%) | 1 (2.0%) |
| HPI | 5.07 | 77.41 | 38.74 | <100 | Low polluted | 64 (100.0%) | 59 (100.0%) | 50 (100.0%) |
| | | | | >100 | High polluted | 0 (0.0%) | 0 (0.0%) | 0 (0.0%) |
| $C_d$ | −5.84 | −2.90 | −5.02 | <1 | Low | 64 (100.0%) | 59 (100.0%) | 50 (100.0%) |
| | | | | 1–3 | Medium | 0 (0.0%) | 0 (0.0%) | 0 (0.0%) |
| | | | | <3 | High | 0 (0.0%) | 0 (0.0%) | 0 (0.0%) |

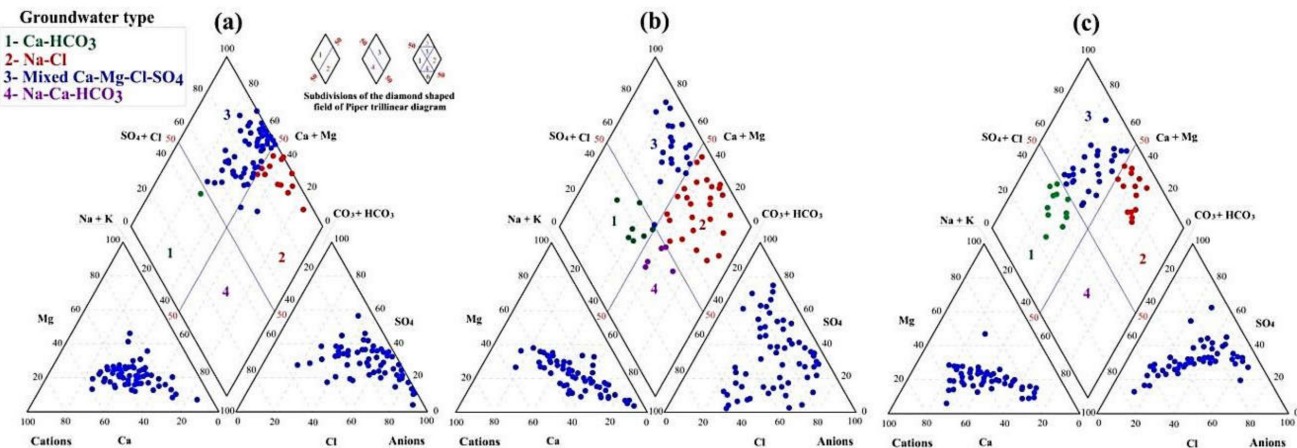

**Figure 6.** Geochemical facies and water type for selected wadies: (**a**) Wadi Marawani, (**b**) Wadi Fatimah, and (**c**) Wadi Qanunah, Makkah Al-Mukarramah Province.

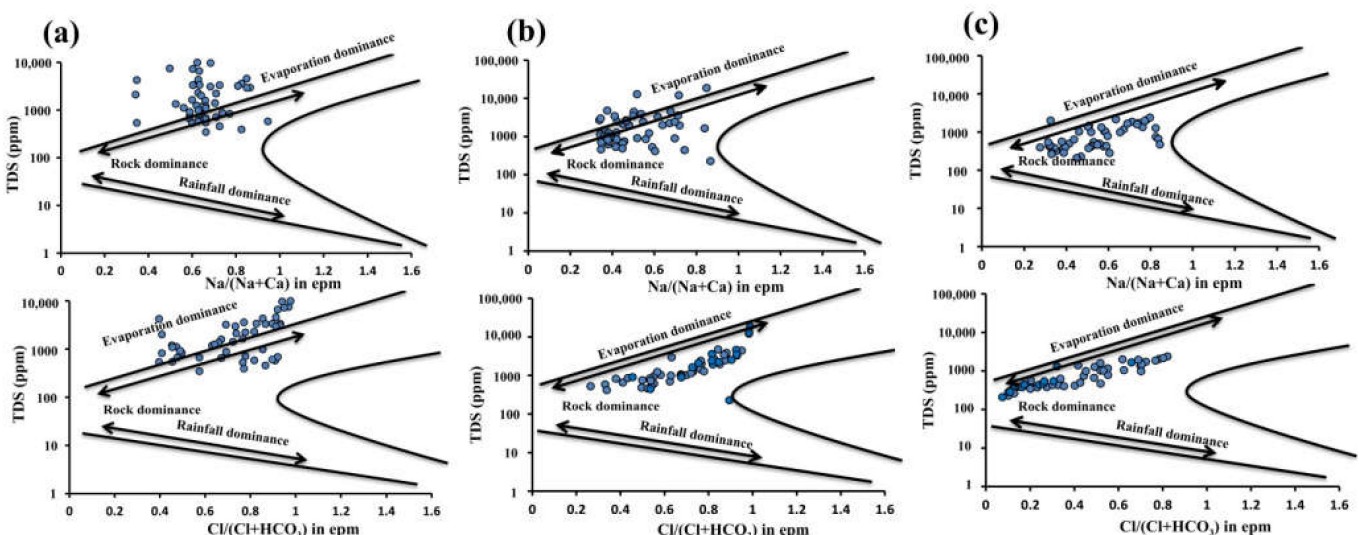

**Figure 7.** Geochemical controlling mechanisms for selected wadies: (**a**) Wadi Marawani, (**b**) Wadi Fatimah, and (**c**) Wadi Qanunah, Makkah Al-Mukarramah Province.

### 3.3.1. Drinking Water Quality Index (DWQI)

According to Equation (1), the WQIs models were applied to evaluate groundwater quality across three study locations, which were classified based on the purity of frequently stately water quality indices. The DWQI across the three wadies was created for assessing

the appropriateness of groundwater for potable usages. Table 6 shows the calculated DWQI values in the acquired groundwater samples, which varied from 22.69 to 545.53, with an average of about 97.11. The DWQI categorization (Table 6) classified roughly 2.5% of groundwater as excellent, 18.0% as good, 28.0% as bad, 21.5% as extremely poor, and 30.0% as unfit for drinking due to the increase of $Na^{2+}$ and $Cl^{-}$ concentrations as a result of seawater intrusion, evaporation, and ion exchange processes. The DWQI distribution map (Figure 8) revealed that most of groundwater samples cannot be used for safe drinking due to the evaporation process, groundwater–rock interaction, reverse ion exchange processes, and seawater intrusion, as well as a low amount of rainfall, especially in the northern wadies of the Makkah region (Marawani and Fatimah). At the same time, in Wadi Qanunah in the south of Makkah, 50% of the samples ranged from excellent to good water for drinking use due to the floods following the rains, which have a positive impact on the recharge of groundwater in the area and its quality.

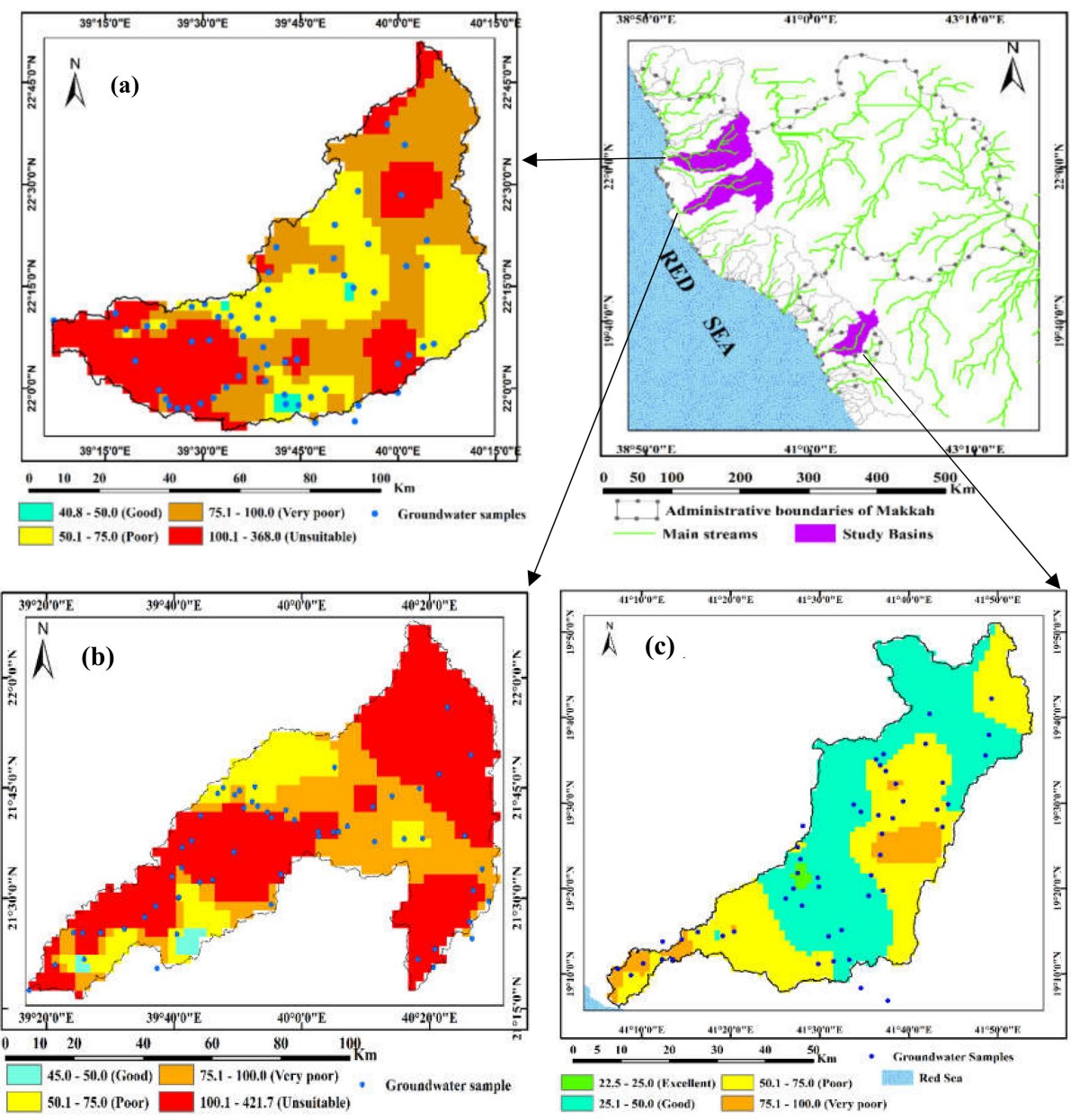

**Figure 8.** Spatial distribution map of DWQI for selected wadies: (**a**) Wadi Marawani, (**b**) Wadi Fatimah, and (**c**) Wadi Qanunah, Makkah Al-Mukarramah Province.

### 3.3.2. Pollution Indices (PIs)

Table 6 and Figure 9 showed statistical descriptions and classifications of groundwater samples with respect to PIs, such as HPI and $C_d$. The HPI readings varied from 5.07 to 77.41, with a mean value of 38.74, indicating that all groundwater points were less contaminated than the threshold HPI value (100). The calculated $C_d$ values for the examined groundwater samples indicated that the $C_d$ levels varied from $-5.84$ to $-2.90$, with a mean value of $-5.02$. The $C_d$ values showed that all groundwater points had negative values ($C_d < 1$), suggesting a minimal degree of pollution and improved metal purity (Figure 10).

The PIs are regarded as an important technique for determining the appropriateness of water for drinking purposes in terms of metals [57]. According to the PIs findings, the majority of groundwater locations in the research region are not suitable for drinking. This might be attributable to a faulty drainage system, seawater intrusion, and rock–water interaction [20]. As a result, groundwater in the research region needs be treated before it can be consumed. The PIs, containing the HPI and $C_d$, demonstrated that groundwater in the wadies was low in metal contamination. According to the categorization of PI values, the data indicated two sets of heavy metal impacts (Table 7 and Figure 11). The obtained PI values demonstrated that metals had no influence on the groundwater points (PI < 1.0), whereas Mn had a slight effect in Wadi Fatimah (PI = 1–2), as shown in Table 7 and Figure 10. The PI results revealed two classes of heavy metal effects based on the classification of PI levels (Table 7 and Figure 11).

**Table 7.** Groundwater quality assessment for the selected wadies in Makkah Al-Mukarramah Province based on metal impacts.

| Metals | PI | | | | | | | | |
| | Wadi Marawani | Class | Effect | Wadi Fatimah | Class | Effect | Wadi Qanunah | Class | Effect |
|---|---|---|---|---|---|---|---|---|---|
| Al | 0.58 | I | No effect | 0.18 | I | No effect | 0.58 | I | No effect |
| Cu | 0.16 | I | No effect | 0.02 | I | No effect | 0.18 | I | No effect |
| Fe | 0.69 | I | No effect | 0.04 | I | No effect | 0.29 | I | No effect |
| Mn | 0.96 | I | No effect | 1.43 | II | Slight effect | 0.54 | I | No effect |
| Ni | 0.56 | I | No effect | 0.43 | I | No effect | 0.41 | I | No effect |
| Zn | 0.12 | I | No effect | 0.02 | I | No effect | 0.04 | I | No effect |

By comparing the geographical distribution maps of DWQI, HPI, and $C_d$ findings, it is evident that the groundwater quality for drinking is deteriorating near the downstream parts of the wadies. Based on the link between the DWQI and PIs, metals had no effect on groundwater quality in the research regions. As a result of differences in heavy metal concentration measurement techniques, the groundwater quality in the study region revealed low levels of water contamination for HPI and $C_d$. As a result, combining the DWQI and PIs provides a beneficial and relevant technique for evaluating groundwater quality for drinking reasons using physiochemical parameters related to heavy metals.

### 3.4. The Variation of Spectral Reflectance under Different Groundwater Quality Levels

The spectrum measurements for 173 groundwater samples collected in the three wadies were obtained. Figure 12 demonstrates the relationship between wavelength and reflectance at various DWQI levels across the three wadies. The results in Figure 12 show that there were distinct differences in respective spectral features associated to reflectance values throughout the VIS and NIR. From 800 to 850 nm, the combined form of the spectral signature and the wavelength locality exhibited a minor variation. The variance in spectra from blue to red (400–700 nm) indicated a higher difference than the NIR range (800–1148 nm). The spectral curve showed high values of reflectance at excellent levels of DWQI and low values of reflectance at unsuitable levels of DWQI. According to the data, the difference in spectral signature between DWQI levels, the VIS, and the NIR area is the

highest among different electromagnetic spectrum regions. The difference in the composite form of the spectral signature produced at different DWQI levels appears to be a result of water component absorption. This conclusion emphasizes the importance of wavelength ranges in the VIS and NIR spectra in analyzing the three wadies' WQIs. The properties of light reflected from water bodies at various wavebands in the magnetic spectrum can be utilized as markers of changes in the water physiochemical composition [36,37,58]. Surprisingly, these alterations result in considerable variations in the SRIs obtained from water samples throughout the whole spectral spectrum at certain bands. Several studies have found that the spectra at VIS, red-edge, and NIR have a stronger relationship with various physicochemical properties of different water bodies than other spectral regions, hinting that these spectral bands may be utilized to evaluate water quality [59–66].

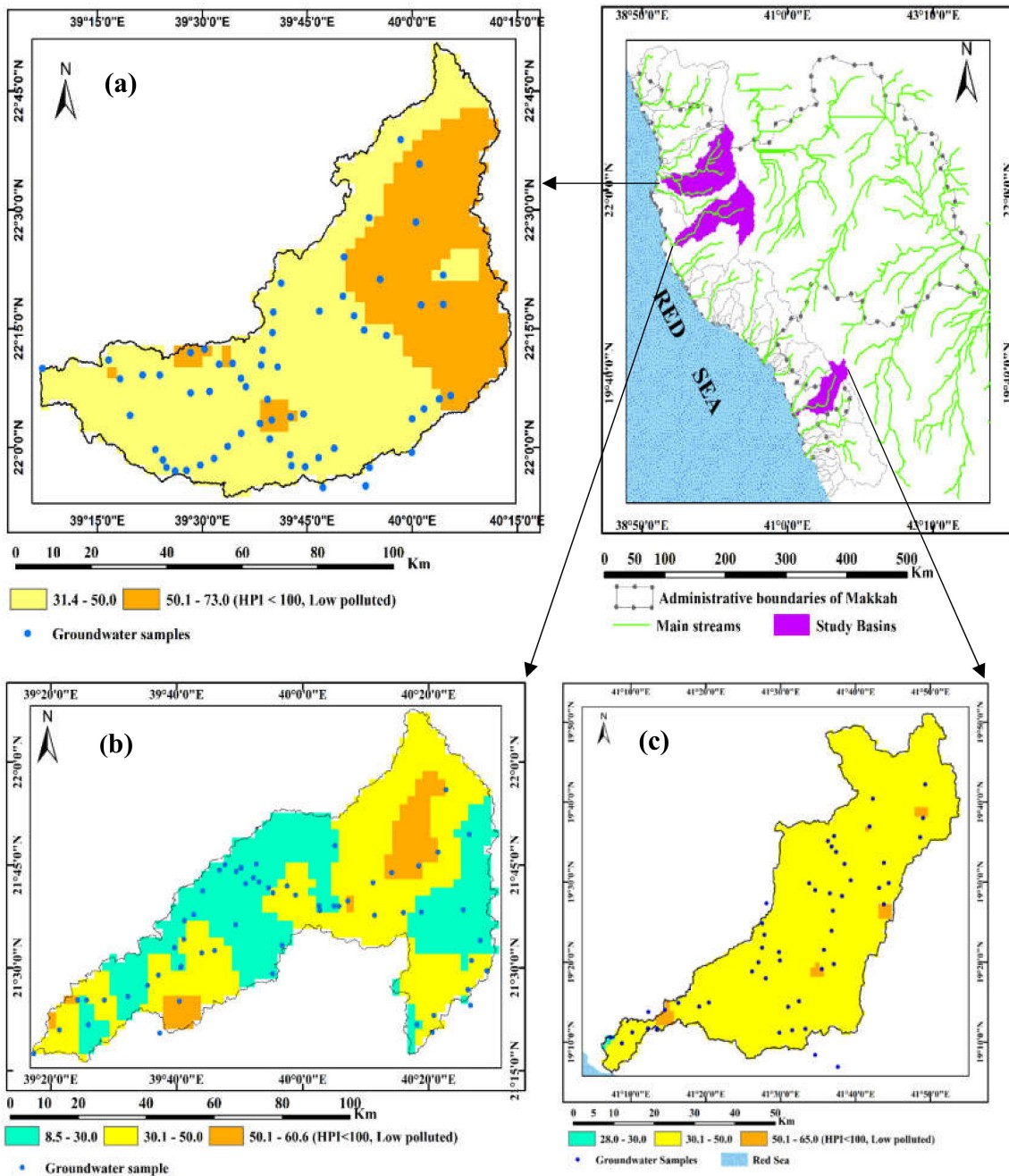

**Figure 9.** Spatial distribution map of HPI for selected wadies: (**a**) Wadi Marawani, (**b**) Wadi Fatimah, and (**c**) Wadi Qanunah, Makkah Al-Mukarramah Province.

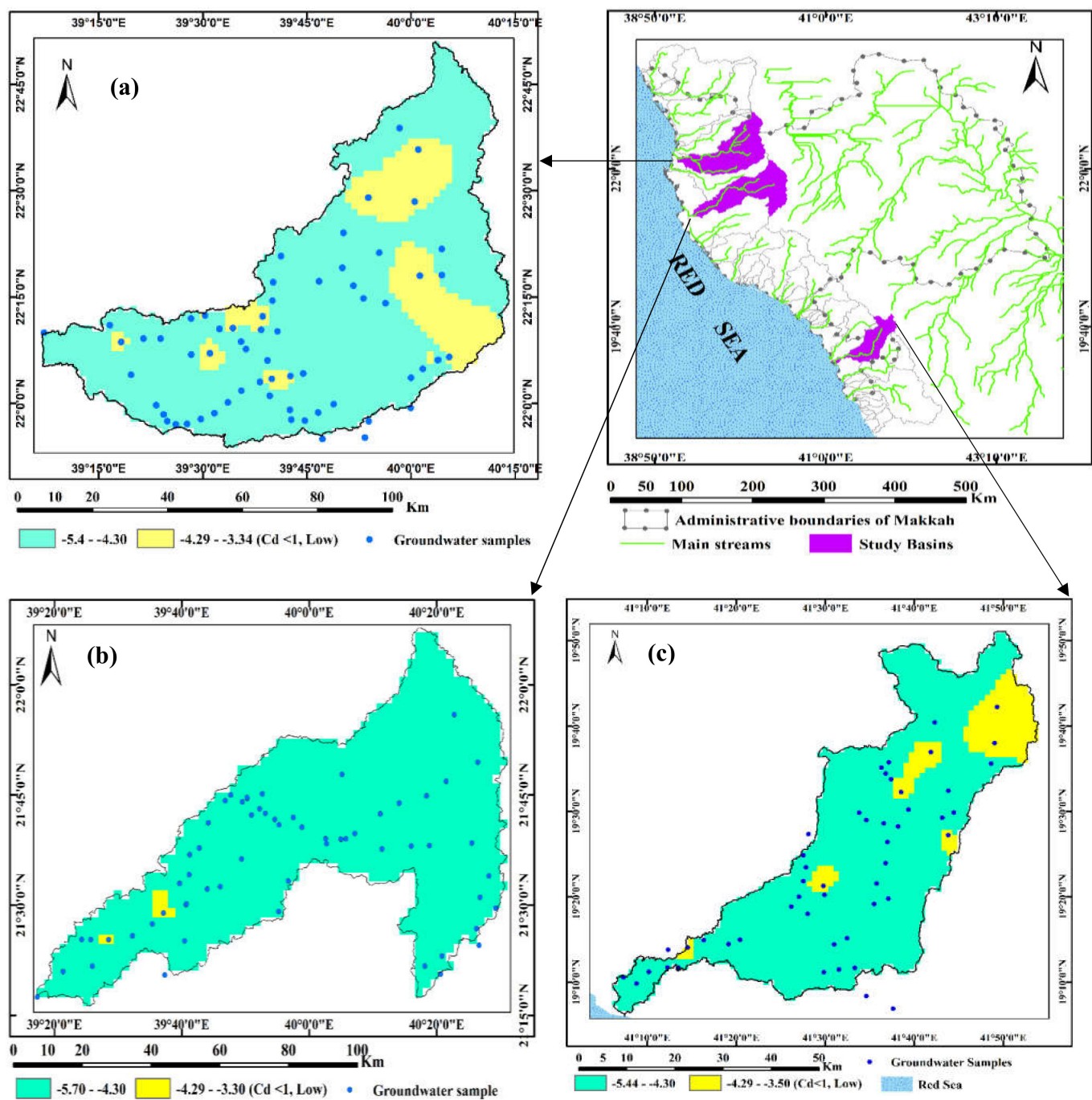

**Figure 10.** Spatial distribution map of $C_d$ for selected wadies: (**a**) Wadi Marawani, (**b**) Wadi Fatimah, and (**c**) Wadi Qanunah, Makkah Al-Mukarramah Province.

### 3.5. Performance of Different SRIs to Estimate Groundwater Quality Indices

Several studies have been carried out in order to investigate the effectiveness of optical remote sensing tools, such as satellite images and airborne and ground-based sensors, to estimate water quality levels. However, the majority of them have only investigated water quality metrics such as TSS, dissolved organic matter (DOM), chlorophyll-a, and turbidity [67–73]. However, a few studies employed nearby hyperspectra to determine the DWQI, HPI, and $C_d$ of water [50]. The two-band slice map was used to pick $R^2$ for the associations between the four WQIs and the SRIs. The hotspot areas, depending on the color scale for the best $R^2$ based on information received from the WQIs in the VIS and NIR regions, were used to identify the best connections between the SRIs and WQIs. The

hotspots (color scale) of the best $R^2$ identified were used to choose SRIs. Table 8 shows the correlation coefficient (R) values for the association between the four measured WQIs and SRIs. Linear regression models demonstrated the significant relationships for the majority of SRIs paired with DWQI: R ranged from (−0.40 to 0.75), (0.44 to 0.76), and (−0.44 to 0.65); with TDS, R ranged from (0.46 to 0.74), (0.45 to 0.76), and (0.41 to 0.65) for Wadi Marawani, Wadi Fatimah, and Wadi Qanunah, respectively. This is due to the presence of different levels of drinking water quality, ranging from the level of excellent to the level of unsuitable for drinking.

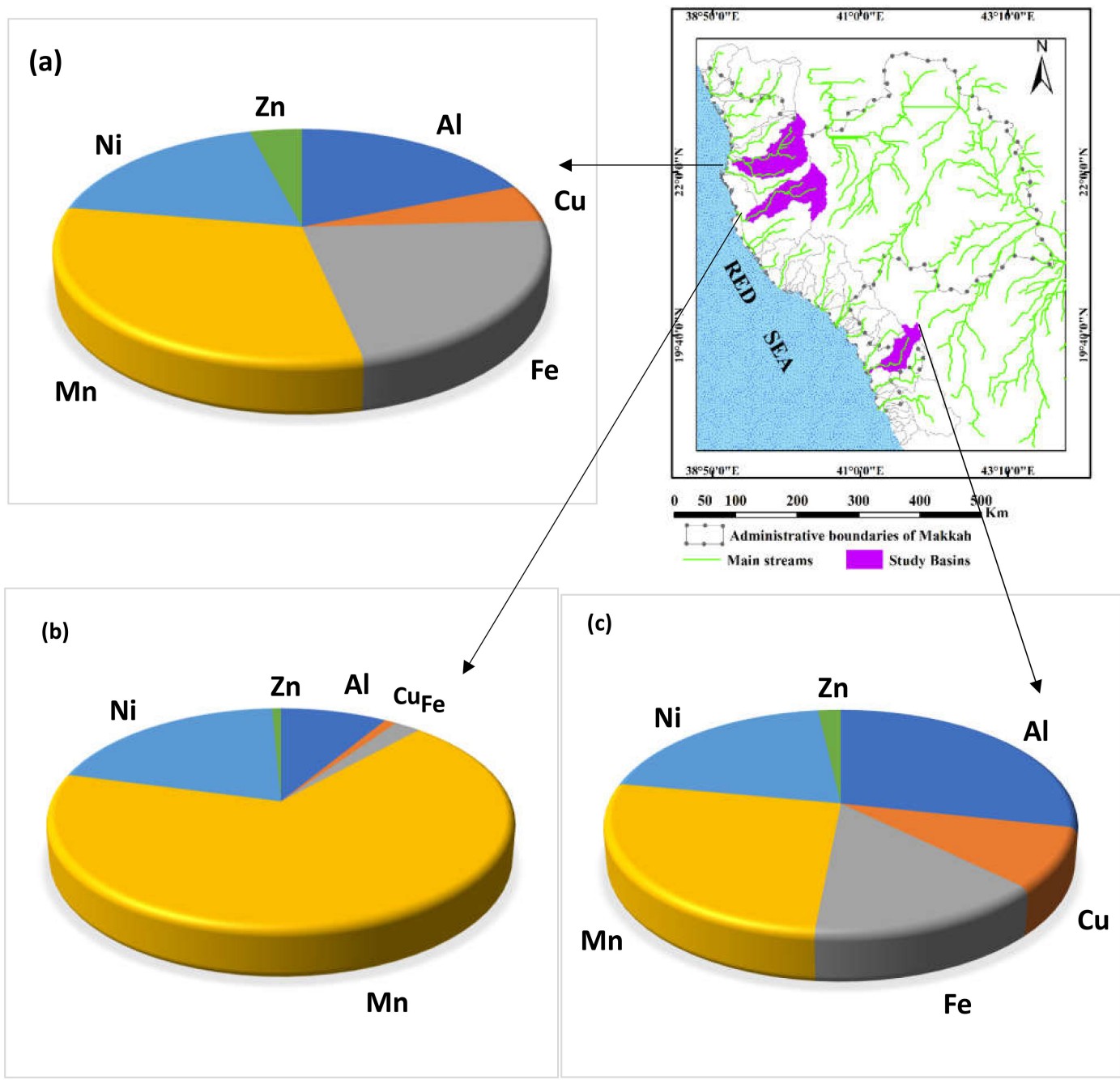

**Figure 11.** Three-dimensional pie showing the relative pollution index of metals for selected wadies: (**a**) Wadi Marawani, (**b**) Wadi Fatimah, and (**c**) Wadi Qanunah, Makkah Al-Mukarramah Province.

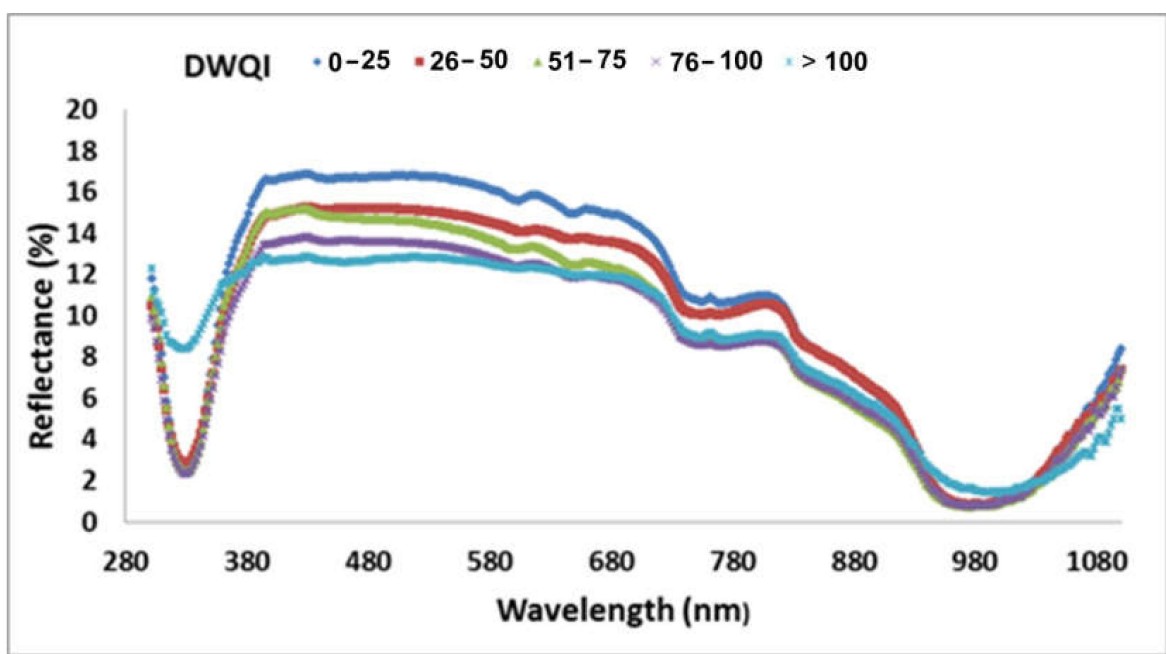

**Figure 12.** Spectra reflectance of the various levels of drinking groundwater quality index across the three wadies samples.

New SRIs presented better performance for assessing the DWQI and TDS for three wadies than the published SRIs in Table 8. SRIs created using NIR and VIS regions, such as $RSI_{1122,590}$, $RSI_{1122,488}$, and $RSI_{1122,488}$, seem to be good indicators to assess DWQI and TDS. Wang et al., (2017) discovered that the DWQI may be easily detected at the peak wavelengths of 700–720 nm and 1070 nm. Spectrum curves for diverse water samples indicated multiple strong high absorbency zones at 700, 750, 950, and 980 nm, as well as low absorbency zones at 452, 703, and 850 nm, according to the researchers. Furthermore, Gad et al. [50] discovered that SRIs derived from the VIS and NIR regions had substantial associations with DWQI for two River Nile branches in Egypt. On the contrary, SRIs failed to assess both HPI and $C_d$, as the results of both indices suggested a minimal degree of pollution and improved metal purity. In addition, Gad et al. [50] found that SRIs could be used to assess metal index (MI) under different pollution levels.

**Table 8.** Correlation coefficient of the linear association between SRIs and DWQI, TDS, HPI, and $C_d$.

| | Wadi Marawani (n = 64) | | | | Wadi Fatimah (n = 59) | | | | Wadi Qanunah (n = 50) | | | |
|---|---|---|---|---|---|---|---|---|---|---|---|---|
| | DWQI | TDS | HPI | $C_d$ | DWQI | TDS | HPI | $C_d$ | DWQI | TDS | HPI | $C_d$ |
| B/R | 0.46 | 0.46 | −0.01 | −0.21 | −0.13 | −0.13 | 0.13 | −0.07 | −0.44 | −0.49 | −0.12 | 0.07 |
| G/R | 0.49 | 0.49 | −0.04 | −0.25 | −0.13 | −0.14 | 0.09 | −0.05 | −0.39 | −0.43 | −0.11 | 0.07 |
| NIR/R | −0.07 | −0.03 | −0.10 | 0.00 | 0.14 | 0.14 | −0.21 | −0.12 | 0.45 | 0.41 | −0.05 | −0.15 |
| NIR/B | −0.40 | −0.38 | −0.03 | 0.17 | 0.11 | 0.11 | −0.11 | −0.02 | 0.50 | 0.52 | 0.09 | −0.09 |
| NIR/G | −0.38 | −0.35 | −0.03 | 0.17 | 0.12 | 0.12 | −0.14 | −0.03 | 0.48 | 0.47 | 0.04 | −0.12 |
| $RSI_{1122,454}$ | 0.60 | 0.57 | −0.25 | −0.24 | 0.73 | 0.73 | 0.18 | 0.33 | 0.65 | 0.65 | 0.09 | −0.04 |
| $RSI_{1122,470}$ | 0.66 | 0.63 | −0.25 | −0.25 | 0.76 | 0.76 | 0.18 | 0.34 | 0.65 | 0.64 | 0.08 | −0.04 |
| $RSI_{1124,472}$ | 0.63 | 0.62 | −0.15 | −0.19 | 0.68 | 0.70 | 0.26 | 0.26 | 0.54 | 0.56 | 0.04 | −0.12 |
| $RSI_{1122,480}$ | 0.69 | 0.66 | −0.24 | −0.26 | 0.75 | 0.75 | 0.18 | 0.33 | 0.65 | 0.63 | 0.08 | −0.04 |
| $RSI_{1122,488}$ | 0.70 | 0.67 | −0.24 | −0.26 | 0.75 | 0.74 | 0.18 | 0.33 | 0.64 | 0.62 | 0.07 | −0.04 |
| $RSI_{1122,510}$ | 0.73 | 0.71 | −0.23 | −0.27 | 0.71 | 0.71 | 0.17 | 0.30 | 0.63 | 0.60 | 0.06 | −0.03 |
| $RSI_{1122,554}$ | 0.75 | 0.74 | −0.20 | −0.28 | 0.61 | 0.60 | 0.15 | 0.24 | 0.60 | 0.56 | 0.03 | −0.03 |
| $RSI_{1124,570}$ | 0.74 | 0.74 | −0.11 | −0.23 | 0.44 | 0.45 | 0.21 | 0.13 | 0.49 | 0.48 | −0.03 | −0.14 |
| $RSI_{1122,590}$ | 0.75 | 0.74 | −0.18 | −0.29 | 0.48 | 0.48 | 0.15 | 0.19 | 0.56 | 0.52 | 0.03 | −0.01 |

Note: All R values higher than −0.40 or 0.40 in table are significant.

### 3.6. Prediction of Different WQIs Using PLSR and PCR Models

SRIs are straightforward techniques for evaluating water quality and may be used to construct spectral sensors that are small and light for monitoring and controlling water quality on a broad scale in a timely and cost-effective way. However, each SRI focuses on just two or three sensitive waveband combinations [36,37]. This makes it challenging to construct effective SRIs for evaluating water quality under a number of potentially perplexing circumstances, such as large differences in water component amounts and types, as well as the impact of these differences on the saturation levels of the water quality measurements [42]. Therefore, in this work, the PLSR and PCR included SRIs as input variables to predict WQIs. Tables 9 and 10 describe the $R^2$ and RMSE, as well as the slope of equations of the two models used as criteria to estimate WQIs in Wadi Marawani, Wadi Fatimah, and Wadi Qanunah in Table 1. Generally, depend on these criteria, the PLSR and PCR models offered a more precise estimate of DWQI and TDS and failed to predict the HPI and $C_d$ in Tables 9 and 10. For example, The PLSR model produced reasonable DWQI and TDS estimations in calibration datasets with $R^2$ of 0.69 and 0.69, 0.80 and 0.84, and 0.41 and 0.40 in Wadi Marawani, Fatimah, and Qanunah, respectively (Table 10). The PLSR model produced reasonable DWQI and TDS estimations in validation datasets with $R^2$ of 0.60 and 0.60, 0.75 and 0.77, and 0.42 and 0.43 in Wadi Marawani, Fatimah, and Qanunah, respectively. The PCR model produced reasonable DWQI and TDS estimations in validation datasets with $R^2$ of 0.58 and 0.55, 0.71 and 0.78, and 0.40 and 0.37 in Wadi Marawani, Fatimah, and Qanunah, respectively (Table 10). In general, the PLSR and PCR models provided better estimation for DWQI and TDS than the indvidual SRI. The LVs ranged from 1 to 7 and were chosen to support the calibration data for the PLSR models of six WQIs without over-fitting through Wadi Marawani, Fatimah, and Qanunah (Table 9). In addition, the LVs ranged from 1 to 9 and were chosen to support the calibration data for the PCR models of six WQIs without over-fitting through Wadi Marawani, Fatimah, and Qanunah (Table 10). The optimal LVs was selected based on the lowest value of RMSE. Table 10 indicated variation in the slope of the linear relationship between measured and predicted values of PLSR models for each index, with TDS showing the highest slope (0.7739) and HPI showing the lowest slope (−0.0598). Furthermore, Table 10 indicated variation in the slope of the linear relationship between measured and predicted values of PCR models for each index, with TDS showing the highest slope (0.7725) and HPI showing the lowest slope (−0.0514). In both PLSR and PCR models, the RMSE value of each index was the smaller in Wadi Fatimah compare to the other two wadies. This is due to the presence of different levels of water quality between the three wadies.

**Table 9.** Outcomes of calibration and validation models of PLSR for the association between observed and predicted values for DWQI, TDS, HPI, and $C_d$ of the three wadies.

| | Variable | Calibration | | | | Validation | | |
|---|---|---|---|---|---|---|---|---|
| | | LVs | $R^2$ | RMSE | Equation | $R^2$ | RMSE | Equation |
| Wadi Marawani | DWQI | 6 | 0.69 *** | 44.52 | y = 0.6882x + 34.49 | 0.58 *** | 52.29 | y = 0.6323x + 41.862 |
| | TDS | 6 | 0.69 *** | 1974.80 | y = 0.6865x + 920.59 | 0.55 *** | 2358.24 | y = 0.631x + 1100.4 |
| | HPI | 1 | 0.05 | 9.01 | y = 0.049x + 42.896 | 0.03 | 9.30 | y = 0.0163x + 44.366 |
| | $C_d$ | 4 | 0.18 * | 0.53 | y = 0.1753x − 3.9717 | 0.07 | 0.57 | y = 0.1292x − 4.2123 |
| Wadi Fatimah | DWQI | 6 | 0.80 *** | 38.74 | y = 0.8041x + 23.252 | 0.75 *** | 46.52 | y = 0.7271x + 31.232 |
| | TDS | 7 | 0.84 *** | 1296.72 | y = 0.8378x + 417.3 | 0.77 *** | 1555.53 | y = 0.7739x + 529.15 |
| | HPI | 3 | 0.15 * | 0.05 | y = 0.1494x + 26.367 | 0.00 | 17.68 | y = 0.0942x + 27.843 |
| | $C_d$ | 1 | 0.10 | 0.44 | y = 0.0998x − 4.8178 | 0.00 | 0.48 | y = 0.0055x − 5.3217 |
| Wadi Qanunah | DWQI | 1 | 0.41 *** | 15.46 | y = 0.4111x + 32.043 | 0.42 *** | 16.00 | y = 0.3842x + 33.516 |
| | TDS | 1 | 0.40 *** | 466.36 | y = 0.3983x + 521.21 | 0.43 *** | 484.43 | y = 0.3689x + 541.47 |
| | HPI | 1 | 0.01 | 7.69 | y = 0.0091x + 39.369 | 0.00 | 8.25 | y = −0.0598x + 42.122 |
| | $C_d$ | 1 | 0.01 | 0.55 | y = 0.0088x − 4.8545 | 0.00 | 0.58 | y = −0.0377x − 5.0867 |

Note: *, *** Statistically significant at $p \leq 0.05$ and at $p \leq 0.001$, respectively.

**Table 10.** Outcomes of calibration and validation models of PCR for the association between observed and predicted values for DWQI, TDS, HPI, and $C_d$ of the three wadies.

| | Variable | Calibration | | | | Validation | | |
|---|---|---|---|---|---|---|---|---|
| | | LVs | $R^2$ | RMSE | Equation | $R^2$ | RMSE | Equation |
| Wadi Marawani | DWQI | 7 | 0.69 *** | 44.26 | y = 0.6918x + 34.082 | 0.60 *** | 51.83 | y = 0.6532x + 37.369 |
| | TDS | 7 | 0.69 *** | 1965.51 | y = 0.6894x + 911.96 | 0.60 *** | 2321.90 | y = 0.6325x + 1028.8 |
| | HPI | 1 | 0.03 | 9.10 | y = 0.0294x + 43.781 | 0.01 | 9.36 | y = 0.0015x + 45.056 |
| | $C_d$ | 1 | 0.08 | 0.56 | y = 0.0828x − 4.4171 | 0.05 | 0.58 | y = 0.0527x − 4.5628 |
| Wadi Fatimah | DWQI | 6 | 0.80 *** | 38.74 | y = 0.8041x + 23.252 | 0.71 *** | 46.70 | y = 0.7034x + 34.53 |
| | TDS | 9 | 0.85 *** | 1249.78 | y = 0.8493x + 387.69 | 0.78 *** | 1530.39 | y = 0.7725x + 481.98 |
| | HPI | 1 | 0.05 | 17.31 | y = 0.0468x + 29.547 | 0.03 | 17.92 | y = 0.0117x + 30.655 |
| | $C_d$ | 1 | 0.02 | 0.45 | y = 0.0186x − 5.2524 | 0.01 | 0.46 | y = 0.0039x − 5.3304 |
| Wadi Qanunah | DWQI | 1 | 0.41 *** | 15.48 | y = 0.4098x + 32.11 | 0.40 *** | 16.04 | y = 0.3819x + 33.581 |
| | TDS | 1 | 0.40 *** | 467.15 | y = 0.3963x + 522.98 | 0.37 *** | 481.46 | y = 0.3794x + 533.92 |
| | HPI | 1 | 0.01 | 7.70 | y = 0.0045x + 39.55 | 0.00 | 8.15 | y = −0.0514x + 41.847 |
| | $C_d$ | 1 | 0.01 | 0.55 | y = 0.0039x − 4.8781 | 0.00 | 0.57 | y = −0.0252x − 5.0232 |

Note: *** Statistically significant at $p \leq 0.001$.

Wang et al. [72] observed that PLSR models with a high number of wavebands outperformed models with only one or two wavebands in predicting inland water quality indices. In addition, Gad et al. [50] found that the PLSR model correctly predicted the DWQI and MI of the two branches of the River Nile in the validation datasets with $R^2$ varying from 0.78 to 0.93, respectively, under different levels of pollution. Again, PLSR and PCR models based on various SRIs may be used in water quality assessment as a unified strategy for remote WQI assessment.

## 4. Conclusions

The quality of groundwater in the Makkah region was investigated in this study to delineate the suitable zones for potable groundwater. The chemical analyses of the groundwater samples indicated that $Na^{2+} > Ca^{2+} > Mg^{2+} > K^+$, $Cl^- > SO_4{}^{2-} > HCO_3{}^- > NO_3{}^- > CO_3{}^{2-}$; and Cu > Fe > Al > Zn > Mn > Ni, respectively. The hydrochemical type of the groundwater samples revealed Ca-HCO₃, Na-Cl, mixed Ca-Mg-Cl-SO₄, and Na-Ca-HCO₃, which is controlled by seawater invasion, weathering process, and groundwater–rock interaction. The different WQIs, such as DWQI, HPI, $C_d$, and PI, were assessed for 173 groundwater samples. According to the DWQI assessment, the overall quality of groundwater samples in the studied areas varied greatly, from excellent (2.5%) to unfit for drinking (30.0%). The HPI, $C_d$, and PI values revealed that all groundwater samples had a low degree of contamination and were not affected by metals, except for Wadi Fatimah, which was slightly affected by Mn due to rock–water interaction. New SRIs derived from NIR and VIS, such as $RSI_{1122,590}$, $RSI_{1122,488}$, and $RSI_{1122,488}$, seem to be good indicators to assess DWQI and TDS. Generally, depending on $R^2$, RMSE, and slope values, the PLSR and PCR were more accurate in determining the DWQI and TDS and failed to predict the HPI and $C_d$. Finally, integrating WQIs, SRIs, PLSR, PCR, and GIS techniques was successful and provided a clear image for evaluating groundwater suitability for drinking and its regulating variables. In the future, the technique described in this work, which combines spectral indices algorithms and PLSR models, should be further assessed to increase its stability under varied groundwater resource conditions.

**Author Contributions:** Conceptualization, M.E.O., M.G., S.E., M.M. and A.A.; fieldwork, A.A., M.E.O. and M.M.; methodology, M.G., S.E., A.A., M.M. and M.E.O.; software, S.E., M.M. and M.G.; validation, S.E., A.A., M.M. and M.E.O.; formal analysis, S.E. and M.E.O.; investigation, M.E.O., A.A. and M.M.; resources, M.E.O., M.M. and A.A.; data curation, M.E.O.; writing—original draft preparation, M.E.O., M.G. and S.E.; writing—review and editing, S.E., M.E.O., M.G. and M.M; supervision, M.E.O.; project administration, M.E.O. All authors have read and agreed to the published version of the manuscript.

**Funding:** This research work was funded by the Deputyship for Research & Innovation, Ministry of Education in Saudi Arabia, under the project number IFPRC–082–123–2020.

**Institutional Review Board Statement:** Not applicable.

**Informed Consent Statement:** Not applicable.

**Data Availability Statement:** All data are provided as tables and figures.

**Acknowledgments:** "The authors extend their appreciation to the Deputyship for Research & Innovation, Ministry of Education in Saudi Arabia for funding this research work through the project number IFPRC–082–123–2020" and King Abdulaziz University, DSR, Jeddah, Saudi Arabia.

**Conflicts of Interest:** The authors declare no conflict of interest.

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
