# Peer review of "Use of Hyperspectral Reflectance and Water Quality Indices to Assess Groundwater Quality for Drinking in Arid Regions, Saudi Arabia"

_water, doi:10.3390/w14152311_

Round 1

Reviewer 1 Report

This is a review of the manuscript entitled “Use of Hyperspectral Reflectance and Water Quality Indices to Assess Groundwater Quality for Drinking in Arid Regions, Saudi Arabia” by Abdulaziz Alqarawy et al. In this manuscript, the authors based on the field sampling collected from three Wadies (Marawani, Fatimah and Qanunah) of  Makkah Al-Mukarramah Province, comprehensive examinations of groundwater chemistry, mechanism analysis, and quality evolution, which have been carried out worldwide, can give strong knowledge for   groundwater security.   However, there are some minor problems in this manuscript. Therefore, the paper is suitable for this journal.

1. Introduction: The logical relationship between the front and the back is chaotic, there are contradictory expressions, and the context is not tightly combined. It is suggested to reorganize the expression logic and strengthen the logical relationship of the content.

2. Abstract: Lack of primary and secondary relationship of content, failed to highlight the main content, it is suggested to further refine and summarize, and clearly state the primary and secondary relationship of research content.

3. The description of groundwater sampling and analysis is confusing, and the description of different methods is not clear. It is suggested that the sampling and analysis should be described in sections, and the description of analysis should be more organized.

4. Fig.1: The format of figure C is not uniform, such as border line and north arrow.

5. Fig.4: add the linear fitting equation and r value.

6. Chapter 2: Different summaries are not all juxtaposed, so it is suggested to adjust their logical relationship.

7. The pictures are not clear enough, including Figures 5, 8, 9 and 10.

8. Lack of expression and analysis of fig. 7.

9. The geographical distribution map in the first paragraph of page 20 refers to that map, and there is a lack of analysis content of this map.

10. The sixth section of the third chapter only describes and summarizes the process of lack of analysis.

Author Response

We greatly appreciate your critical observations as well as your constructive and helpful comments. We hope that we could address your questions/comments by the explanations and revisions made in the manuscript. We believe that the manuscript is substantially improved after making the suggested revisions.

Reviewer 2 Report

The manuscript is very interesting, the authors have described a groundwater quality of three Wadies in Saudi Arabia and their possibility of use for the needs of public water supply.

To achieve better quality of the manuscript, I suggest the following:

Chapter 2.1.

Location map is missing. All toponyms mentioned in the manuscript must be shown on the map.

·        Line 138-146: The location is visible in Figure 1. I suggest removing that part and increasing the letters on Figure 1 to make it more visible. 

A better hydrogeological and geological description of each separate Wadi is needed. The geometry of the aquifer is missing, also changes in groundwater levels and quality during different hydrological conditions.

·        The polygon of the administrative area in Figure 1a and 2a and 2b is not the same. Need to harmonize.

Chapter 2.2.

·        Were samples taken from boreholes? From what depth and under what hydrological condition?

Chapter 2.5.

·        Table 3 shows the arithmetic means of the values of the quality parameters? At first, it is obvious that they do not correspond to table 6?

Chapter 3.1.

·        EC is usually in units of µS/cm or mS/cm. Here it is µS/cm, I assume, but in the text and in the table it is shown as s/cm. sec/cm?

Chapter 3.3.

·        Line 489: Just a possibility? Needs better explanation.

Chapter 3.4.

·        Line 562: The sentence sounds like a part is missing.

Author Response

(The authors gave the same response as above.)

Reviewer 3 Report

1. Use unique abbreviation for WQI, i think the study is not focused on Irrigation water quality index(IWQI). so i recommended to use the unique abbreviation 

2. TDS is the water quality parameter, how did you add it in the pollution index method. If the TDS is major concern you should add it in separate

3. For K and Na, electron count is different than HCO3, use unique representation method 

4. GIS is a tool like excel, Matlab. I am sure that software are tool to represent the result and it not use for identify the contamination history in the study area

5. Add the equation for Charge balance errors (CBE)

6. Typing error in line 247. Correct it

7. What is the purpose of adding table.2. It is repeated in text. Remove it 

8. Na2+ or + ? Line 421. Correct it 

9. Redraw Fig.7. it is not clear. rock water interaction, precipitation and evaporation limits were changes. Check and correct it 

10. In Table.7 It shows WQI, in section 3.3.1 DWQI. Correct it 

11.  Use common range in Table.7 and fig.8 Range 25-50 is good(Table.7) and 40-50 is good (fig.8). correct it 

12. Improve Fig.8 quality and follow reclassification method in GIS tool, improve the boundary thickness and reduce the pixel size limit during IDW interpolation 

13. Carry out the correction for fig.9 and 10.

14. Which parameter is highly influence  the DWQI value, what is the reason for 30% of sample were unfit. There is no clear discussion.

15. I recommended few recent study to enhance the quality of MS, 

https://doi.org/10.1007/s10653-022-01237-5; https://doi.org/10.1007/s12517-022-09553-x; https://doi.org/10.1007/s13201-021-01525-y; https://doi.org/10.1007/s13201-021-01522-1 

Author Response

(The authors gave the same response as above.)

Author Response

We greatly appreciate your critical observations as well as your constructive and helpful comments. We hope that we could address your questions/comments by the explanations and revisions made in the manuscript. We believe that the manuscript is substantially improved after making the suggested revisions

Round 2

Reviewer 2 Report

On the Figure 1 it could not be seen some toponyms mentioned in the manuscript (Riyadh, Al-Madinah Al-Munawarah, Al Bahaand Asir). Suggestion to the authors to put the toponyms on the figure 1.

In the manuscript the authors added just a depth of the aquifers, but not the geological composition along the depth of the aquifer. Suggestion to the authors to include more detailed geological description of the each aquifer. 

It cannot be seen on the figure 1 or 2 any change of the administrative area. Polygons of the Province are not harmonized.

In the chapter 2.2. there is not any added data about the depths from the samples were taken and also about the hydrological condition in that moment.

If you look to the measured value of pH in the table 2 (7.07) and compare it to table 5 it can be concluded that the sample is taken in the Wadi Fatimah and that is not the arithmetic mean of the samples taken in all Wadies. Also, in the line 385-386 is stated that the DWQI is derived by the concentrations of parameters for each sample site. That is not corresponding to the table 2.

Now the unit of EC is µs/cm. Has to be µS/cm.

Author Response

(The authors gave the same response as above.)

Reviewer 3 Report

Accept

Author Response

We greatly appreciate your critical observations as well as your accepting the modification carried out on the Manuscript

Round 3

Reviewer 2 Report

1-      It cannot be still seen on the figure 1 or 2 any change of the administrative area. Polygons of the Province are not harmonized. Coordinates on the fig. 1c and 1, 8 and 8c, 9 and 9c, 11 are too small. Has to be increased.

2-    In the chapter 2.2. There is not any added data about the depths from the samples were taken and also about the hydrological condition in that moment. As it could be seen, there is no any changes in the chapter 2.2. even the authors in their response claimed different.

Author Response

We greatly appreciate your critical observations as well as your constructive and helpful comments. We hope that we could address your questions/comments by the explanations and revisions made in the manuscript. We believe that the manuscript is substantially improved after making the suggested revisions (Round 3).
